# Stepwise assembly of the human replicative polymerase holoenzyme

**Mark Hedglin\*, Senthil K Perumal, Zhenxin Hu, Stephen Benkovic\***

Department of Chemistry, The Pennsylvania State University, University Park, United States

**Abstract** In most organisms, clamp loaders catalyze both the loading of sliding clamps onto DNA and their removal. How these opposing activities are regulated during assembly of the DNA polymerase holoenzyme remains unknown. By utilizing FRET to monitor protein-DNA interactions, we examined assembly of the human holoenzyme. The results indicate that assembly proceeds in a stepwise manner. The clamp loader (RFC) loads a sliding clamp (PCNA) onto a primer/template junction but remains transiently bound to the DNA. Unable to slide away, PCNA re-engages with RFC and is unloaded. In the presence of polymerase (polδ), loaded PCNA is captured from DNA-bound RFC which subsequently dissociates, leaving behind the holoenzyme. These studies suggest that the unloading activity of RFC maximizes the utilization of PCNA by inhibiting the build-up of free PCNA on DNA in the absence of polymerase and recycling limited PCNA to keep up with ongoing replication.

## Introduction

Replicative DNA polymerases (pols) alone are distributive, synthesizing very few nucleotides of complementary DNA before disengaging from an elongating primer strand. To achieve the high processivity required for efficient DNA replication, replicative pols anchor to ring-shaped sliding clamps, forming the holoenzyme. Sliding clamps are dimers or trimers of identical subunits aligned head-to-tail, forming a highly-conserved, toroidal structure with two structurally distinct faces and a central cavity large enough to encircle double-stranded DNA and slide freely along it. The 'C-terminal face' contains all C-termini and serves as a platform for interaction with replicative pols (*Hedglin et al., in press*; *Yao and O'Donnell, 2012*). In solution, most sliding clamps are closed, requiring an enzyme-catalyzed mechanism which; (1) disrupts an interface within the sliding clamp ring and holds it open for assembly; (2) targets it to primer/template (P/T) junctions where DNA synthesis is initiated; (3) orients it correctly for interaction with pols; and (4) closes it around DNA (*Yao et al., 1996*; *Schurtenberger et al., 1998*; *Matsumiya et al., 2003*; *Zhuang et al., 2006*; *Paschall et al., 2011*). Such feats are achieved by heteropentameric complexes referred to as clamp loaders, which utilize ATP to catalyze site-directed loading of sliding clamps onto DNA.

Clamp loaders are composed of two to five distinct subunits that assemble into a tight heteropentameric complex with an overall structure that is conserved throughout evolution. Each subunit belongs to the AAA+ superfamily of ATPases that contain characteristic ATP-binding/hydrolysis motifs that convert the energy from ATP binding and hydrolysis to mechanical force. Over the years, information gathered from detailed mechanistic and structural studies has converged on a similar sequential mechanism for clamp loader-mediated assembly of the DNA pol holoenzyme (*Hedglin et al., in press*; *Yao and O'Donnell, 2012*). In the presence of ATP, clamp loaders bind the C-terminal face of their respective clamp and open it for assembly. Once formed, the open clamp•clamp loader complex binds a P/T junction, adopting a 'notched screw cap arrangement' that matches the helical geometry of the DNA duplex (*Simonetta et al., 2009*; *Kelch et al., 2011*). In this orientation, ATP hydrolysis is optimized and

**\*For correspondence:**
muh218@psu.edu (MH);
sjb1@psu.edu (SB)

**Competing interests:** The authors declare that no competing interests exist.

**Reviewing editor**: John Kuriyan, University of California, Berkeley, United States

**eLife digest** All living organisms use enzymes called replicative DNA polymerases to produce copies of their genome during cell division. These enzymes promote the formation of new DNA strands by catalyzing the polymerization of deoxyribonucleotides, the single units that make up DNA. In humans these polymerases contain multiple protein subunits, as well as a specific cofactor—a magnesium ion—that is required for the enzyme to be active.

DNA polymerases are able to add several nucleotides at once because they are anchored to ring-shaped protein complexes called sliding clamps that encircle the DNA template. This structure, known as the holoenzyme, is able to slide freely along the DNA template, which allows the polymerase to promote the addition of nucleotides in a highly efficient manner.

Protein complexes called clamp loaders are responsible for attaching the holoenzyme to the DNA template, and also for detaching it. Studies of model organisms, including bacteria, viruses and yeast, have provided insights into the assembly of the holoenzyme in humans, but the exact mechanism behind this process has remained unknown.

Now, Hedglin et al. use fluorescence resonance energy transfer (FRET), a powerful microscopy technique that can monitor interactions between proteins, and also between proteins and DNA, to study the assembly of the holoenzyme. Whenever a sliding clamp is loaded onto a DNA template in the absence of polymerase, the clamp loaders quickly remove it. Whenever a polymerase is present, however, it captures the sliding clamps; the clamp loaders then dissociate from the newly assembled holoenzyme and DNA replication begins.

By revealing that clamp loaders recycle scarce sliding clamps, and that they boost the efficiency of holoenzyme assembly by preventing clamps from accumulating on DNA in the absence of polymerase, Hedglin et al. have redefined our understanding of human holoenzyme assembly.

the clamp loader reverts to a low-affinity DNA-binding state upon hydrolysis of ATP and ejects from the P/T junction, leaving behind the loaded clamp. Concurrent or subsequent to clamp loader ejection, the open sliding clamp ring contracts around the P/T junction and is ready to anchor an incoming pol to the 3′ end of the primer strand, forming the holoenzyme (*Hedglin et al., in press*; *Yao and O'Donnell, 2012*).

Support for the generic model described above was primarily garnered from studies focused on holoenzyme assembly within *E. coli*, T4 bacteriophage, and *S. cerevisiae* while parallel human studies are gravely lacking. Also, distinguishing subtleties are present within each of these model systems such that a defined model for assembly of the human pol holoenzyme cannot be inferred. For instance, the fate of the clamp loader upon loading the clamp onto DNA is quite distinct for each model system. In T4 bacteriophage, the clamp loader chaperones the incoming pol onto the correct face of loaded clamp prior to dissociating from the clamp•DNA complex (*Trakselis et al., 2003*; *Perumal et al., 2012*). In *E. coli,* the clamp loader disengages from the P/T junction but remains at the replication fork through association with the replicative helicase and single-stranded DNA binding protein and chaperones one of the three pols bound to its own subunits to the newly loaded clamp (*Downey and McHenry, 2010*). In *S. cerevisiae*, a recent report suggests that the clamp loader does not actually eject from the P/T junction but rather remains part of the pol holoenzyme through association with one of the sliding clamp subunits (*Kumar et al., 2010*). Furthermore, the few studies on assembly of the human pol holoenzyme failed to account for the clamp unloading activity of the clamp loader, replication factor C (RFC). In a replicating cell, the number of P/T junctions are in excess of sliding clamps, necessitating an efficient unloading mechanism for recycling during S-phase (*Yao et al., 1996*; *Leu et al., 2000*). The human sliding clamp, proliferating cell nuclear antigen (PCNA), is very stable on DNA by itself and is catalytically unloaded by RFC in an ATP-dependent manner. Such an activity is also present in archaea and bacteria while it seems to be missing from *S. cerevisiae* (*Yao et al., 1996*; *Cai et al., 1997*; *Cann et al., 2001*; *Matsumiya et al., 2003*; *Kumar et al., 2010*). In T4 bacteriophage, enzyme-catalyzed clamp removal is unnecessary since the gp45 sliding clamp is unstable on DNA in the absence of polymerase (*Kaboord and Benkovic, 1996*; *Soumillion et al., 1998*; *Perumal et al., 2012*).

Currently, holoenzyme assembly within humans has only been inferred from studies focused on the clamp loading-unloading pathway. Such studies addressed only one direction of the pathway at a time.

The forward direction (clamp loading) was monitored beginning with only free components (DNA, PCNA, RFC) while the reverse direction (clamp unloading) was monitored by subjecting isolated PCNA•DNA complexes to RFC. Thus, it is assumed that RFC ejects immediately upon loading sliding clamps onto DNA, before the arrival of the polymerase, and then rebinds to unload them only after the polymerase disengages. However, without monitoring clamp loading and unloading simultaneously, the temporal correlation between the opposing activities of RFC and how they are regulated during assembly of the DNA polymerase holoenzyme remains unknown.

Here we report studies on the assembly of the human replicative polδ holoenzyme. By utilizing FRET signals from fluorescently labeled P/T DNA and PCNA to monitor protein-DNA interactions, we were able to observe both activities of RFC in real time and directly monitor assembly of the DNA pol holoenzyme. We obtained data detailing the step-wise manner in which the human polδ holoenzyme is assembled; (1) RFC loads a stoichiometric amount of PCNA onto a P/T junction in an ATP-dependent manner; (2) upon hydrolysis of ATP, RFC releases the closed clamp ring onto DNA but remains transiently bound near the P/T junction; (3) unable to 'slide away,' PCNA re-engages with bound RFC in the absence of polδ and together RFC and all of the loaded PCNA dissociate from the DNA; (4) in the presence of polδ, DNA-bound PCNA is captured and physically occluded from the unloading activity of RFC and then; (5) RFC dissociates from the P/T junction, leaving behind the functional holoenzyme consisting of PCNA and polδ. Thus, human RFC does not immediately eject from P/T DNA in between PCNA loading and unloading as has been assumed for so long and the replicative pols play a paramount role by capturing loaded PCNA from DNA-bound RFC to inhibit unloading of PCNA and complete assembly of the holoenzyme. Upon dissociation of polδ from PCNA and DNA, RFC may then unload the clamp for recycling. These studies provide the first comprehensive analysis of assembly of the human replicative pol holoenzyme. The outcomes of these studies suggest that the unloading activity of the human clamp loader serves a dual purpose; it increases the efficiency of holoenzyme assembly by inhibiting the build-up of free clamps on DNA in the absence of replication and recycles scarce clamps to keep up with ongoing DNA replication.

## Results

### RFC loads PCNA onto primer-template DNA in the presence of ATP

In order to study holoenzyme assembly within humans, we utilized FRET to monitor protein-DNA interactions. A forked DNA substrate in agreement with the minimal requirements for assembly of human RFC and PCNA onto DNA was labeled with a Cy3 dye that served as the FRET donor (*Tsurimoto and Stillman, 1991*). This substrate, referred to herein as Cy3-P/T DNA, resembles the in vivo replication fork and also carries a 3′-biotin label (*Figure 1A* and *Table 1*). Together with the flap, the 3′-biotin label in complex with Neutravidin prevents loaded clamp from sliding off the DNA (*Yao et al., 2000*). The position of PCNA residue 107 within the clamp ring is structurally conserved among eukaryotes (*Figure 1B*) and has been previously utilized for labeling as a FRET acceptor through mutation to cysteine (*Gulbis et al., 1996*; *Zhuang et al., 2006*; *Kumar et al., 2010*). A mutant PCNA with the surface-exposed cysteines mutated and asparagine 107 changed to cysteine (N107C) was labeled with Cy5-maleimide (*Figure 1—figure supplement 1*). The labeled mutant PCNA (Cy5-PCNA) stimulated the ATPase activity of RFC equal to that for wild-type PCNA, indicating a fully functional protein (*Figure 1—figure supplement 2*). A truncated form of RFC (hRFCp140ΔN555) was used in these studies in which the N-terminal 555 amino acids of the large subunit (RFC1, p140) were deleted to remove the nonspecific DNA-binding affinity of RFC (*Figure 1—figure supplement 1*). This region of the human clamp loader is conserved among eukaryotes and is dispensable for PCNA loading (*Uhlmann et al., 1997*; *Podust et al., 1998*; *Gomes et al., 2000*). Herein, the truncated form of the clamp loader (hRFCp140ΔN555) will be referred to as simply RFC.

The Cy3-P/T DNA substrate was tested for PCNA loading by monitoring the steady-state FRET signal between the Cy3 and Cy5 dyes by fluorescence spectroscopy (*Figure 1C*). In the presence of RFC and ATP (or ATPγS), a distinct FRET signal was observed at 665 nm, indicating that loading of Cy5-PCNA onto Cy3-P/T DNA is favored at equilibrium (*Figure 1D*). Both RFC and ATP (or ATPγS) are required for PCNA loading, confirming that the ATP-dependent PCNA-loading activity of RFC is responsible for the observed steady-state FRET signal. In order to study the kinetics of PCNA loading in real time, we pre-assembled the RFC•Cy5-PCNA complex in the presence of ATP and mixed it with Cy3-P/T DNA in a stopped-flow instrument (*Figure 2A*). In the absence of RFC, the signal remains flat

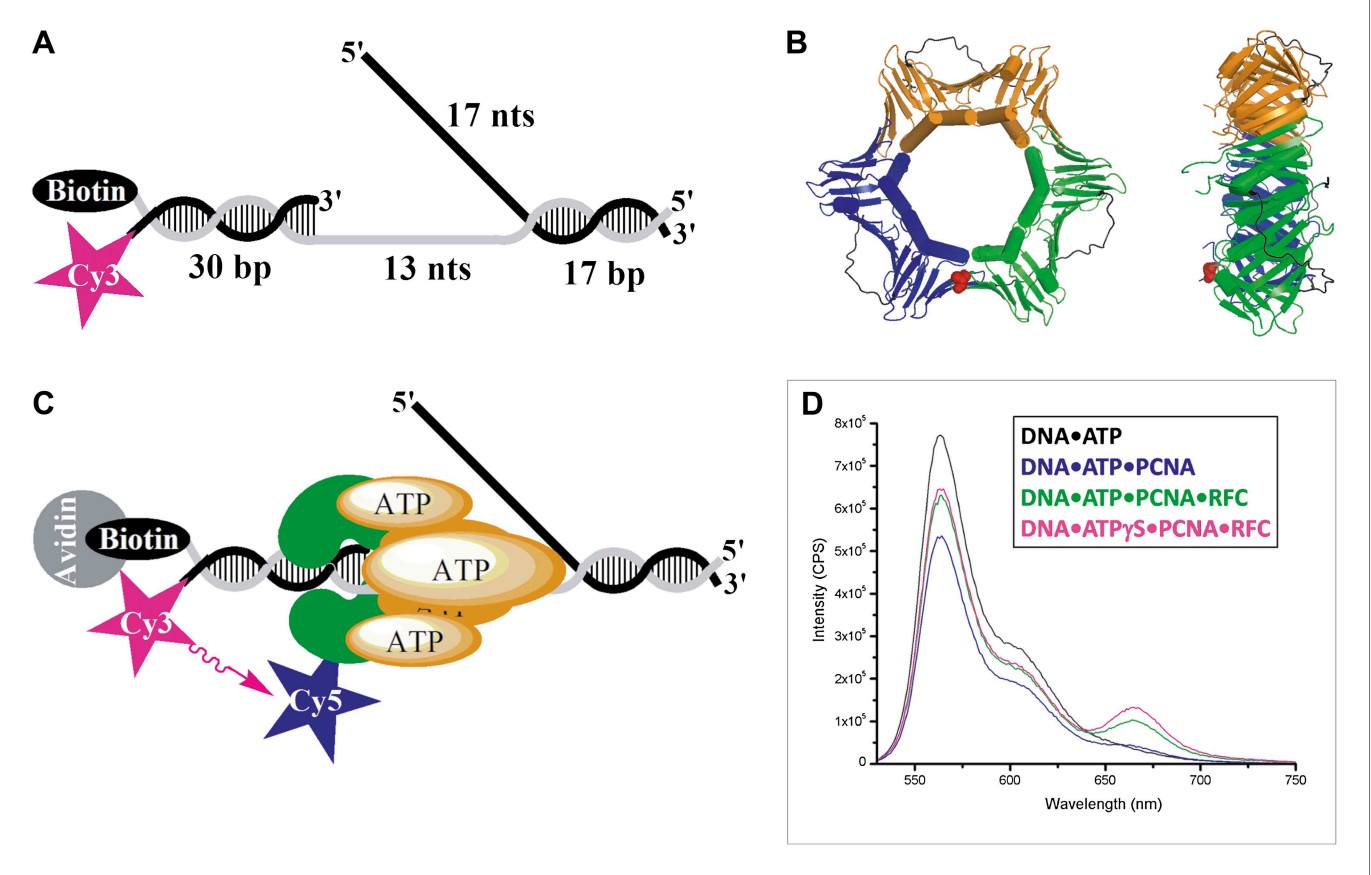

**Figure 1**. RFC-mediated loading of PCNA onto DNA. (**A**) Schematic representation of the Cy3-P/T DNA substrate used in this study. The primer had a Cy3 dye at the 5′ end and biotin was attached to the 3′ end of the template. Sequences for the primer, template, and flap constructs of all substrates used in this study are shown in **Table 1**. The recombinant human proteins used in this study are shown in **Figure 1—figure supplement 1** .(**B**) Model of human PCNA generated using Pymol from PDB code 1AXC (**Gulbis et al., 1996**). PCNA subunits are shown in ribbon form in green, orange, and blue. The asparagine 107 residue, shown in red for one PCNA monomer in space-filling form, was mutated to cysteine for dye labeling. On average, each PCNA trimer has at least one labeled monomer. The mutations nor the labeling of PCNA had any effect on its ability to interact of RFC (**Figure 1—figure supplement 2**). Frontal and side views are shown. (**C**) Schematic representation of RFC-catalyzed loading of PCNA onto DNA. The N107C residue of PCNA is located on the face opposite that which interacts with RFC and faces the Cy3 FRET donor on the P/T DNA. (**D**) Fluorescence emission spectra obtained by exciting the Cy3-P/T DNA with a 514-nm light source. Cy5-PCNA can be excited through FRET from Cy3 only when the two dyes are in close proximity (<~10 nm). Cy5 fluorescence intensity peaks at 665 nm.

The following figure supplements are available for figure 1:

**Figure supplement 1**. SDS–PAGE analysis of recombinant human proteins used in this study.

**Figure supplement 2**. Activities of labeled PCNA mutants in stimulating human RFC ATPase activity.

over the entire time course (**Figure 2B**, black trace) again demonstrating that RFC is responsible for the observed FRET signal. In the presence of RFC, an extended time course of the FRET trace displayed three distinct phases (**Figure 2B**, blue trace). A very rapid increase in FRET occurred within the 'dead time' of the instrument and was not observed in the trace (please see 'Materials and methods'). This is most likely binding of the RFC•ATP•Cy5-PCNA complex to DNA. This was followed by a FRET increase with a rate constant of $k_{inc}$ = 6.4 ± 0.28 s$^{-1}$ and a FRET decrease with a rate constant of $k_{dec,1}$ = 1.4 ± 0.038 s$^{-1}$. After ~5 s, the FRET signal stabilized and remained flat for up to 1 min (**Figure 2B**, Inset).

The rate constants for the two observed phases ($k_{inc}$, $k_{dec,1}$) were both independent of RFC concentration (**Figure 2—figure supplement 1**). This suggests that each observed phase represents a first-order kinetic process, that is, a conformational change. The bimolecular association of the

**Table 1.** DNA constructs for forked DNA substrates used in this study

| Substrate | Template | Primer | Flap |
|---|---|---|---|
| Cy3-P/T DNA | CGG ACT GCA CGT GCC GCG TGG GCA TTC GTC GCG CAG GCT CAG CGT CCA TCG CGA GAC CAG /3Bio/ | /5Cy3/CTG GTC TCG CGA TGG ACG CTG AGC CTG CGC | CGT GGT GGT AGG TGA GGG CGG CAC GTG CAG TCC G |
| Cy3-P/T DNA-no flap | CGG ACT GCA CGT GCC GCG TGG GCA TTC GTC GCG CAG GCT CAG CGT CCA TCG CGA GAC CAG /3Bio/ | /5Cy3/CTG GTC TCG CGA TGG ACG CTG AGC CTG CGC | GCG GCA CGT GCA GTC CG |
| Cy3-P/T DNA-25 bp | CGG ACT GCA CGT GCC GCG TGG GCA TTC GTC GCG CAG GCT CAG CGT CCA TCG CGA G/3Bio/ | /5Cy3/CT CGC GAT GGA CGC TGA GCC TGC GC | CGT GGT GGT AGG TGA GGG CGG CAC GTG CAG TCC G |

RFC•ATP•Cy5-PCNA complex with Cy3-P/T DNA is apparently fast as it was not observed at any RFC concentration on the time scale of our experiments. Interestingly, the overall amplitude ($A_T$) of the signal (i.e., the sum of the amplitudes of all phases in the FRET trace, please see 'Materials and methods') increases linearly throughout the entire range of RFC concentrations and does not saturate at a concentration where the RFC•Cy5-PCNA•Cy3-P/T DNA complex is stoichiometric (*Figure 2D*). In the presence of ATP, binding of human RFC to PCNA is very tight with a $K_D$ ~ 0.2 nM (*Cai et al., 1997*; *Shiomi et al., 2000*). Thus, at 200 nM Cy5-PCNA in *Figure 2B–C*, the RFC•Cy5-PCNA complex is saturated. This suggests that the FRET value at infinite time does not reflect a static endpoint but rather the amount of PCNA loaded onto DNA at equilibrium.

Only the rate constant for the observed FRET increase ($k_{inc}$) was independent of [ATP] (*Figure 2—figure supplement 2A*). Even at the lowest [ATP] (62.5 μM), $k_{inc}$ was maximal, demonstrating that RFC has moderately high affinity for ATP. In the presence of 1 mM ATPγS, $k_{inc}$ splits into two phases, both of which are more than 30-fold slower than $k_{inc}$ observed at 1 mM ATP (*Figure 2—figure supplement 2C*). Due to the differing kinetic characteristics of the FRET increases observed in the presence of ATP and ATPγS, it cannot be concluded that each represents an approach to the same FRET state. Furthermore, the presence of multiple phases within the observed FRET increase with ATPγS may suggest either multistep loading or the presence of more than one population of the ternary complex. Nonetheless, these studies demonstrate that only ATP binding, not hydrolysis, is required for opening of the clamp ring and assembly of the RFC•PCNA•DNA complex, in agreement with previous reports on the human system (*Lee and Hurwitz, 1990*; *Mossi et al., 1997*; *Shiomi et al., 2000*). The rate constant (*Figure 2—figure supplement 2B*) as well as the amplitude (data not shown) for the observed FRET decrease ($k_{dec,1}$) was inhibited at concentrations of ATP greater than 1 mM. Perhaps at [ATP] > 1 mM, nucleotide exchange begins to compete with ATP hydrolysis and/or ADP binding within one or more of the RFC subunits, returning the RFC•ADP complex to the ATP-bound form. In contrast to $k_{inc}$, these results suggest that ATP hydrolysis is required for the observed FRET decrease, $k_{dec,1}$. Indeed, a FRET decrease is not observed in the presence of 1 mM ATPγS (*Figure 2—figure supplement 2C*), a non-hydrolysable analog of ATP.

Collectively, the studies in this section show that PCNA is loaded onto P/T DNA in a process that requires both ATP and RFC. Furthermore, it appears that PCNA loading proceeds through two observed phases; a conformational change that requires only ATP binding followed by a conformational change that requires ATP hydrolysis. The bimolecular association of the RFC•ATP•Cy5-PCNA complex with Cy3-P/T DNA is rapid and was not observed on the timescale of our experiments.

## In the absence of polymerase, RFC loads PCNA and unloads all loaded PCNA within a single binding encounter with P/T DNA

The presence of multiple FRET changes within the traces presented in *Figure 2B–C* may suggest multistep loading. However, under these conditions, it cannot be concluded that each step occurs within the initial binding encounter of RFC•ATP•PCNA and P/T DNA. Furthermore, even though PCNA and RFC•ATP form a tight complex ($K_D$ ~ 0.2 nM), the alternative possibility that multiple reaction species are present cannot be ruled out (*Cai et al., 1997*; *Shiomi et al., 2000*). In order to address this, we carried out pulse-chase experiments as described in *Figure 3A*. A pre-formed RFC•ATP•Cy5-PCNA complex was briefly incubated with Cy3-P/T DNA ('Pulse') and then mixed with a large excess

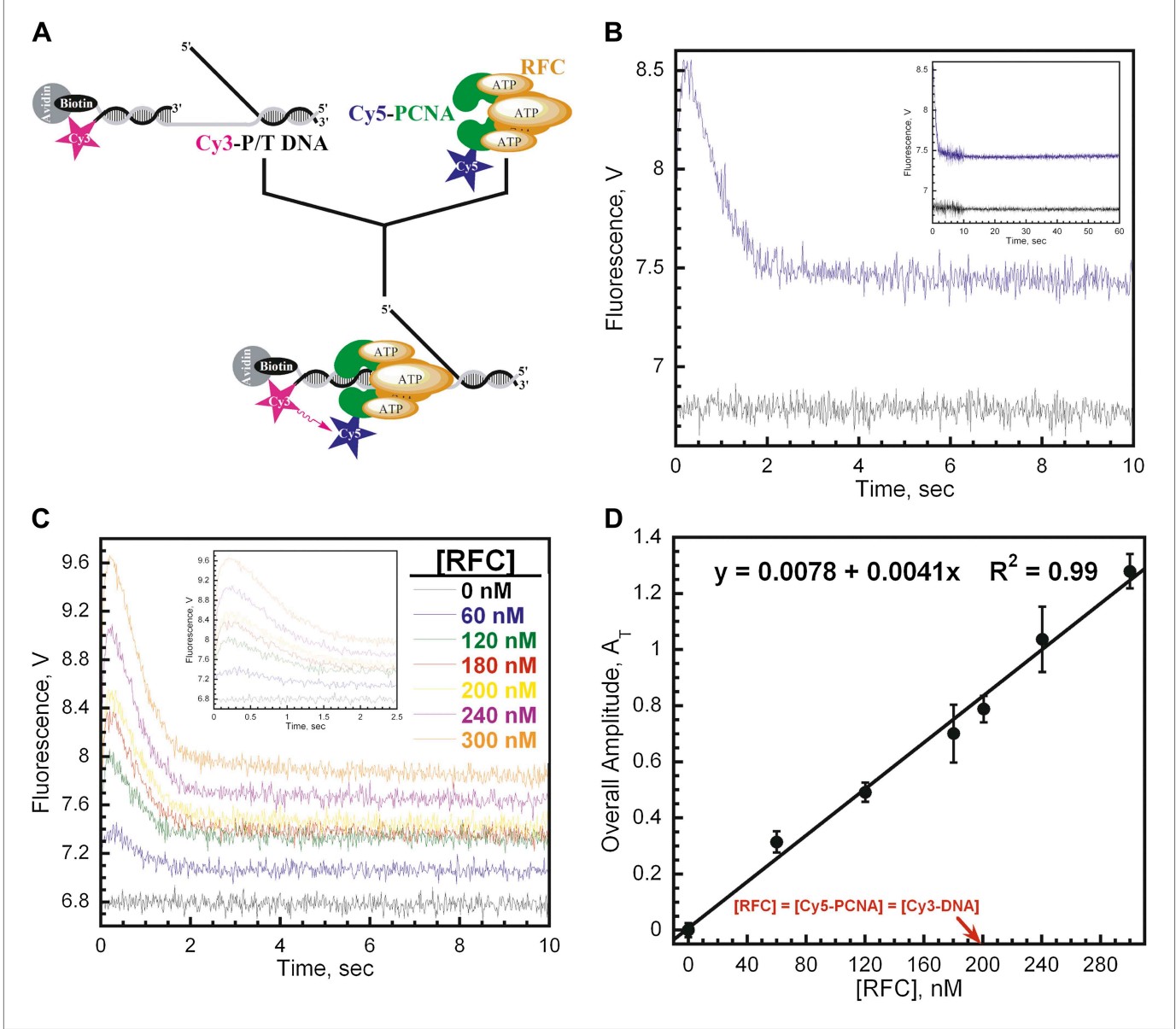

**Figure 2**. RFC loads PCNA with rates independent of RFC concentration. (**A**) Schematic representation of experimental procedure for **Figure 2B**. (**B**) Cy5-PCNA (200 nM) was incubated with RFC (200 nM) in the presence of 1 mM ATP. This preformed RFC•Cy5-PCNA•ATP complex was mixed with Cy3-P/T DNA (200 nM) in a stopped-flow instrument and the FRET signal was followed (Blue trace). The loading traces were fit to a double-exponential equation. If RFC was omitted, no FRET signal was observed (Black trace). An extended time course of 60 s is shown in the inset. (**C**) The experiments depicted in **Figure 2A** were also performed with varying concentrations of either RFC (0–300 nM) or ATP (0–5 mM). Shown in panel C is the RFC titration of the FRET signal. The initial 2.5 s of the FRET traces is shown in the inset. The loading traces were fit to double-exponential equations and the respective rate constants for the fitted FRET increases and decreases are presented as a function of [RFC] in **Figure 2—figure supplement 1**. The ATP titration of the FRET signal is presented in **Figure 2—figure supplement 2**. (**D**) The overall amplitude of the signal (sum of all amplitudes) from **Figure 2C** plotted against [RFC].

The following figure supplements are available for figure 2:

**Figure supplement 1**. RFC loads PCNA with rates independent of RFC concentration.

**Figure supplement 2**. RFC-catalyzed loading of PCNA is dependent upon ATP.

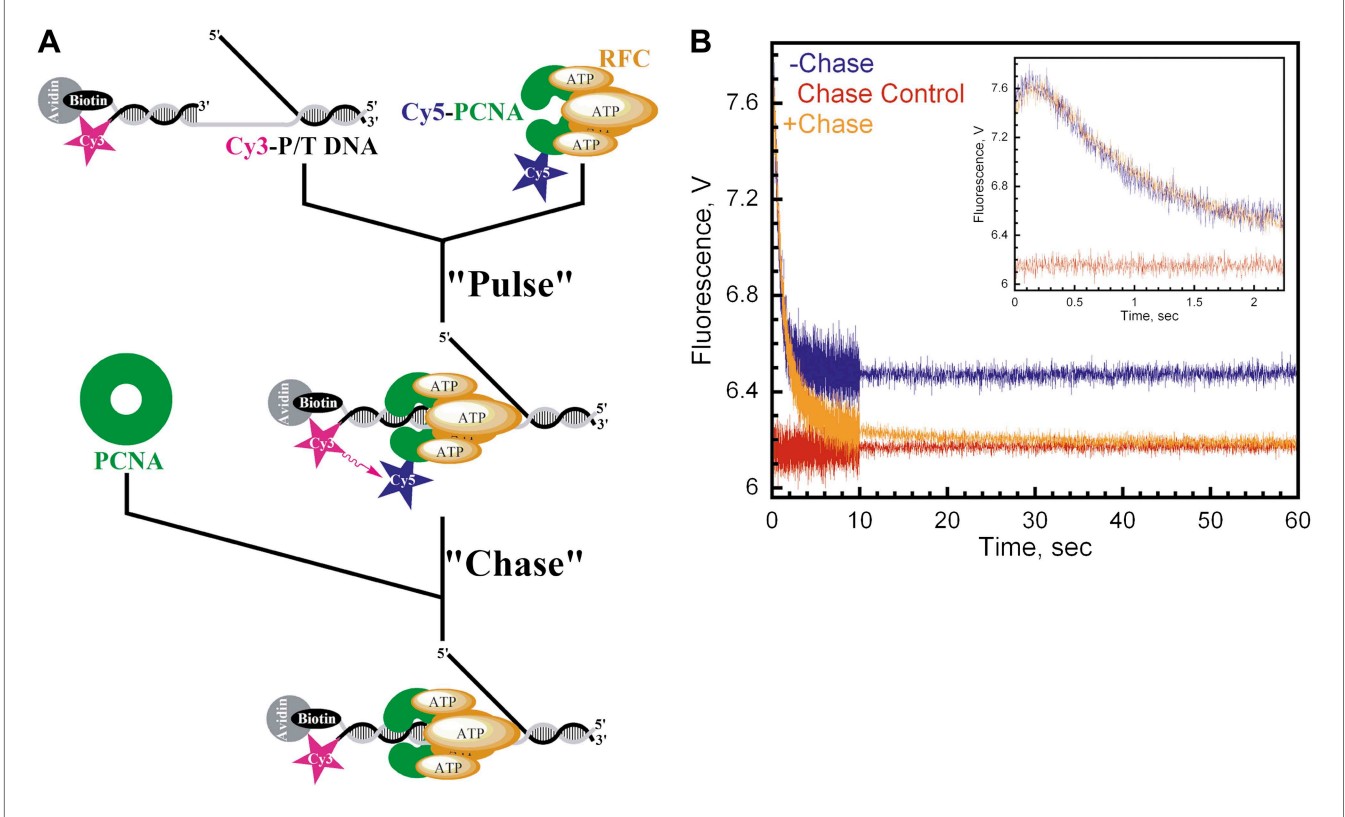

**Figure 3.** Monitoring a single binding encounter between PCNA and P/T DNA. (**A**) Schematic representation of pulse-chase experiment for *Figure 3B*. (**B**) Cy5-labeled PCNA (100 nM) was incubated with RFC (100 nM) and ATP (1 mM). This preformed RFC•Cy5-PCNA•ATP complex was mixed with Cy3-P/T DNA (100 nM) for 37 ± 2.9 ms (Pulse) prior to mixing with unlabeled PCNA at 0 or 2 µM concentration (Chase) in a stopped-flow instrument and the FRET signal was followed. In the absence of chase ('−Chase,' blue trace), the traces appear biphasic as in *Figure 2*. In the presence of chase ('+Chase,' orange trace), two additional phases appear in the FRET decrease. No FRET signal appears if unlabeled trap PCNA (2 µM) is added prior to Cy5-PCNA ('Chase Control,' red trace). The initial 2.25 s of the FRET traces are shown in inset.

unlabeled PCNA ('Chase') in a stopped-flow instrument. Under these conditions, only Cy5-PCNA bound to RFC prior to the addition of Cy3-P/T DNA and chase should generate a FRET signal, thus eliminating free RFC and free Cy5-PCNA. In the absence of chase, the loading traces appear as in *Figure 2* with rate constants of 7.6 ± 0.20 s$^{-1}$ and 1.50 ± 0.083 s$^{-1}$ for the fitted FRET increase ($k_{inc}$) and decrease ($k_{dec,1}$), respectively (*Figure 3B*, blue trace). If unlabeled PCNA chase is added prior to Cy5-PCNA, the FRET signal remains flat (*Figure 3B*, red trace). This represents the zero FRET state and demonstrates that unlabeled PCNA is an efficient trap. When the experiment was carried out in the presence of chase, a FRET increase was observed with a rate constant of $k_{inc}$ = 6.1 ± 0.59 s$^{-1}$, followed by a unique FRET decrease with three distinct exponential decay phases instead of one; $k_{dec,1}$ = 1.6 ± 0.33 s$^{-1}$, $k_{dec,2}$ = 0.43 ± 0.070 s$^{-1}$, and $k_{dec,3}$ = 0.058 ± 5.7 × 10$^{-4}$ s$^{-1}$ (*Figure 3B*, orange trace). The rate constants for the FRET increase ($k_{inc}$) and the initial phase of the FRET decrease ($k_{dec,1}$) agree very well with that observed in the absence of chase, demonstrating that these two observed phases occur within the initial binding encounter of Cy5-PCNA and Cy3-P/T DNA. Furthermore, these two observed phases exactly align in the presence and absence of chase (*Figure 3B*, inset), suggesting that the FRET signal was generated from a single reaction species and that all of the pre-formed RFC•ATP•Cy5-PCNA complex commits through the loading pathway. Thus, dissociation of the RFC•ATP•Cy5-PCNA complex from the Cy3-P/T DNA is much slower (≥10-fold) than the first conformational change ($k_{inc}$). The two additional phases within the FRET decrease only appeared when rebinding of Cy5-PCNA to Cy3-P/T DNA was inhibited by excess unlabeled PCNA chase and decreased the FRET signal to zero where all Cy3-P/T DNA is devoid of Cy5-PCNA. This suggests that these previously unobserved phases represent dissociation of Cy5-PCNA from Cy3-P/T DNA and that the final amplitude observed in the

absence of chase does not reflect a static endpoint but rather the amount of PCNA loaded onto DNA at equilibrium where the rates for PCNA loading and dissociation are equal. Collectively, these studies demonstrate that a single turnover of RFC-catalyzed loading of PCNA actually proceeds through four observed phases, the last two representing dissociation of Cy5-PCNA from Cy3-P/T DNA.

Isolated human PCNA•DNA complexes are incredibly stable in the absence of free DNA ends, with reported half-lives of 22–80 min (*Jonsson et al., 1995*; *Podust et al., 1995*; *Yao et al., 1996*). However, the results presented in *Figure 3* suggest that DNA-bound PCNA is very unstable and rapidly dissociates into solution with a minimal half-life of $t_{1/2} = \ln2/k_{dec,3} = 12$ s. Thus, the appearance of two additional phases ($k_{dec,2}$ and $k_{dec,3}$) within the FRET decrease in the presence of chase (*Figure 3B*) are quite profound as they raise the possibility that upon loading PCNA onto a P/T junction, RFC unloads PCNA without first dissociating into solution. Thus, the two opposing activities of RFC may occur within the same binding encounter with DNA. In order to address this, we directly monitored dissociation of PCNA from P/T DNA.

As depicted in *Figure 4A*, the RFC•Cy5-PCNA•Cy3-P/T DNA complex was pre-assembled in the presence of ATP and then mixed with excess unlabeled PCNA chase. In the absence of chase, the FRET signal does not change over time, again signifying that the loading reaction has reached equilibrium where PCNA loading and dissociation are equal and a net change in the FRET signal is no longer observed (*Figure 4B*, black trace). In the presence of chase, reloading of free Cy5-PCNA is inhibited and the FRET signal decreases with two distinct exponential decay phases (*Figure 4B*, orange trace) with rate constants of $0.40 \pm 0.083$ s$^{-1}$ for the fast phase and $0.039 \pm 0.012$ s$^{-1}$ for the slow phase with relative amplitudes of $83 \pm 1.1\%$ and $17 \pm 1.1\%$ respectively (*Table 2*). These rate constants, along with their relative amplitudes, are independent of the unlabeled PCNA trap concentration (*Figure 4—figure supplement 1*) demonstrating that unlabeled PCNA serves as a passive trap. Furthermore, these values are in excellent agreement with data from *Figure 3B* obtained in the presence of chase. Together, this demonstrates that PCNA indeed rapidly dissociates from P/T DNA in a biphasic manner and suggests that at least two populations of the ternary complex are formed.

Only the faster of the two phases ($k_{dec,2}$) observed in the dissociation of Cy5-PCNA from Cy3-P/T DNA is dependent upon the concentration of ATP (*Figure 4—figure supplement 2B*). Up to 1 mM ATP, $k_{dec,2}$ remains constant at $0.54 \pm 0.080$ s$^{-1}$ and then decreases at concentrations greater than 1 mM ATP. This dependence is reminiscent of that observed for $k_{dec,1}$ (*Figure 2—figure supplement 2*). Furthermore, at each [ATP], $k_{dec,2}$ was 1.5- to 3.4-fold slower than $k_{dec,1}$, demonstrating that each is a distinct ATP hydrolysis-dependent step in the clamp loading-unloading pathway. The slower of the two phases in *Figure 4B* ($k_{dec,3}$) remains constant at $0.057 \pm 0.0044$ s$^{-1}$ across the entire range of [ATP] (*Figure 4—figure supplement 2B*).

Interestingly, spontaneous disassembly of PCNA from P/T DNA through dissociation of the PCNA trimer into its subunits ($6.30 \times 10^{-3} \pm 1.7 \times 10^{-3}$ s$^{-1}$, Senthil Perumal, manuscript in preparation) was never observed in the presence of ATP. This process, referred to as subunit exchange, sets the maximum timescale for disassembly of sliding clamps from DNA in the absence of an enzymatic unloading activity and free DNA ends. In T4 bacteriophage, this process is the sole mechanism for disassembly of the gp45 clamp from DNA (*Kaboord and Benkovic, 1996*; *Yao et al., 1996*; *Soumillion et al., 1998*) and also contributes substantially to disassembly of PCNA from DNA within *S. cerevisiae* (*Kumar et al., 2010*). However, within the human system, this process is ~6.7- and 67-fold slower than the two phases observed in *Figure 4B* and is only observed in ternary complexes formed in the presence of ATPγS (*Figure 4—figure supplement 2A*). Assuming that the two kinetic phases observed in the dissociation of PCNA from P/T DNA represent defined FRET complexes, these results suggest that one or both populations of loaded PCNA may be actively unloaded by (and, hence, dissociate with) RFC. In order to address this, we sought to establish whether a population of RFC dissociates along with either PCNA species and whether or not any PCNA is left behind on P/T DNA.

We carried out the trapping experiments depicted in *Figure 4C* for direct comparison to the PCNA dissociation experiments described in *Figure 4A* above. First, the RFC•ATP•PCNA•Cy3-P/T DNA complex was pre-assembled in the presence of excess Cy3-P/T DNA to ensure all RFC was engaged with PCNA and Cy3-P/T DNA. The PCNA in this complex is unlabeled so a FRET signal is not observed prior to mixing. This complex was then mixed with Cy5-PCNA at a 1:1 ratio of labeled to unlabeled PCNA. Under these conditions, the only source of RFC is from the pre-assembled RFC•ATP•PCNA•Cy3-P/T DNA complex. Thus, if all PCNA is actively unloaded from Cy3-P/T DNA by RFC and rapidly exchanges with Cy5-PCNA, the rate of Cy5-PCNA loading will be entirely rate-limited by dissociation

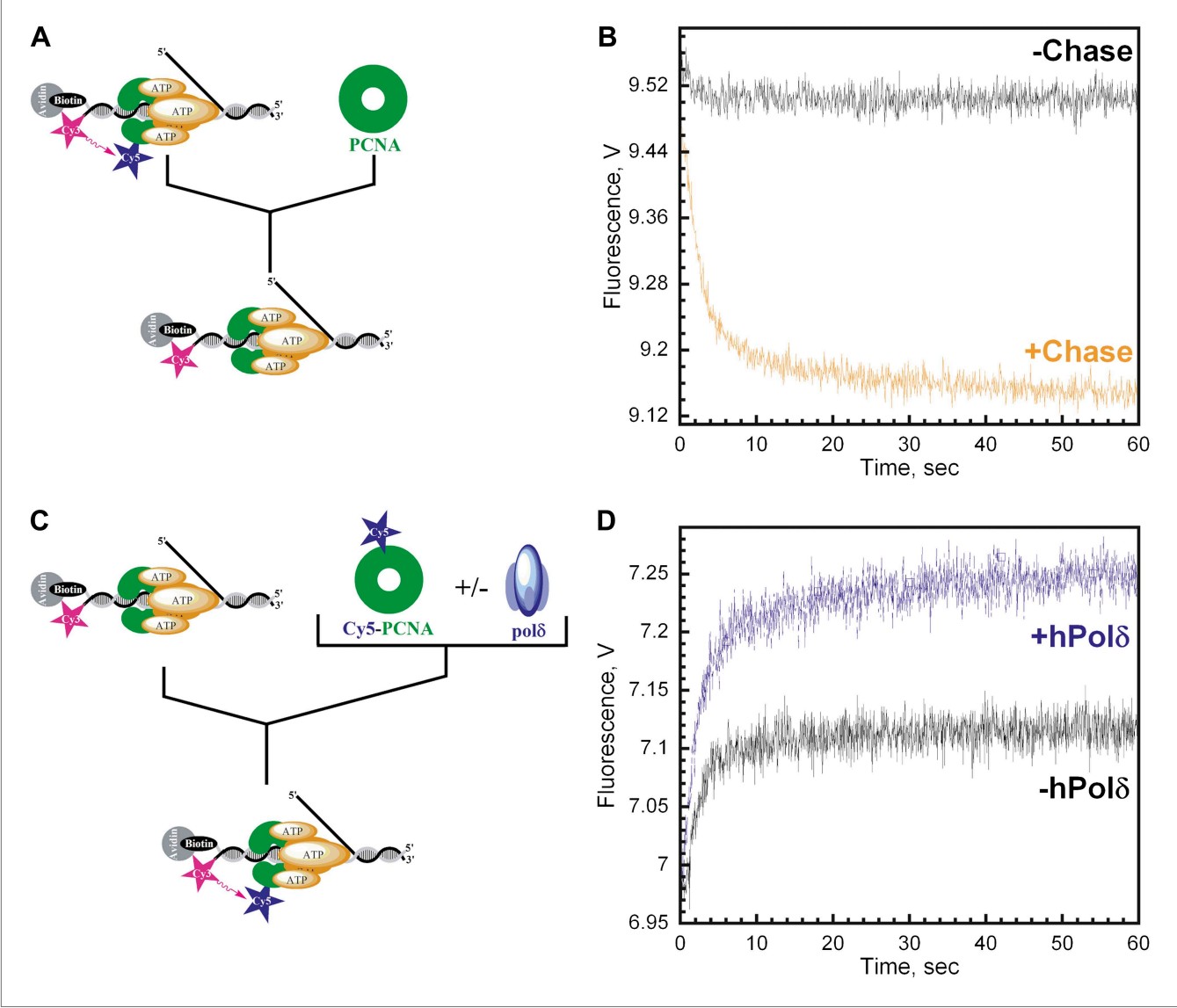

**Figure 4**. polδ inhibits RFC-catalyzed unloading of PCNA to form the holoenzyme. (**A**) Schematic representation of experimental procedure for ***Figure 4B***. (**B**) Cy5-labeled PCNA (100 nM) loaded onto DNA (200 nM) by RFC (100 nM) in the presence of 1 mM ATP was mixed with 0 (−Chase, black trace) or 2 μM unlabeled PCNA chase (+Chase, orange trace) and the FRET signal was followed. The unloading traces were fit to a double-exponential equation and the rate constants and their relative amplitudes are reported in ***Table 2***. These rate constants, along with their relative amplitudes, are independent of the unlabeled PCNA chase concentration (***Figure 4—figure supplement 1***). Only the faster of the two rate phases is dependent upon the concentration of ATP (***Figure 4—figure supplement 2B***). (**C**) Schematic representation of experimental procedure for ***Figure 4D***. (**D**) Unlabeled PCNA (100 nM) loaded onto Cy3-P/T DNA (200 nM) by RFC (100 nM) in the presence of 1 mM ATP was mixed with Cy5- PCNA (100 nM) in the absence (black trace) or presence of polδ (blue trace) and the FRET signal was followed. The loading traces were fit to a double-exponential equation and the rate constants and their relative amplitudes are reported in ***Table 3***. These experiments were also carried out with either varying concentrations of Cy5-PCNA in the absence of polδ (***Figure 4—figure supplement 3***) or with varying concentrations of polδ at a constant [Cy5-PCNA] (***Figure 4—figure supplement 5***).

The following figure supplements are available for figure 4:

**Figure supplement 1**. Unlabeled PCNA serves as a passive trap.

**Figure supplement 2**. Unloading of PCNA by RFC is dependent upon ATP.

**Figure supplement 3**. All PCNA loaded onto P/T DNA dissociates with RFC.

*Figure 4. Continued on next page*

*Figure 4. Continued*

**Figure supplement 4**. The single-stranded DNA flap has no effect on the dissociation of RFC or PCNA from Cy3-P/T DNA.

**Figure supplement 5**. polδ captures a stoichiometric amount of loaded PCNA from DNA-bound RFC to form the holoenzyme.

**Figure supplement 6**. polδ passively captures loaded PCNA from DNA-bound RFC to form the holoenzyme and RFC dissociates from P/T DNA independently of polδ and PCNA.

**Figure supplement 7**. RFC releases PCNA onto P/T DNA.

**Figure supplement 8**. Removing 5 bp from the double-stranded region of the P/T junction has no effect on RFC-catalyzed unloading of PCNA from Cy3-P/T DNA or formation of the polδ holoenzyme.

of the RFC•PCNA complex from Cy3-P/T DNA. Hence, the rate of appearance of the FRET signal should be the same as the rates observed for the disappearance of the FRET signal in *Figure 4B*. Furthermore, the overall amplitude of the FRET increase ($A_{T,Reload}$) should be exactly one-half the overall amplitude for the FRET decrease in *Figure 4B* ($A_{T,Unload}$) if all PCNA is unloaded from Cy3-P/T DNA by RFC, as follows. Upon dissociation of RFC from the pre-assembled complex, the probability that RFC will exchange for and load Cy5-PCNA will be equal to the fraction of Cy5-PCNA in solution. At a 1:1 ratio of PCNA:Cy5-PCNA, the concentrations of unlabeled and labeled PCNA in solution will be equal if RFC unloads all loaded PCNA from the Cy3-P/T DNA and RFC will have an equal probability of loading either onto Cy3-P/T DNA. Thus, the overall amplitude of the FRET increase ($A_{T,Reload}$) will be exactly half the overall amplitude for the FRET decrease ($A_{T,Unload}$) observed in *Figure 4B*. In other words, $A_{T,Reload}/A_{T,Unload}$ will be equal to 0.5, the fraction of labeled PCNA in the reaction. As presented in *Figure 4D*, the FRET time trace was essentially the mirror image of *Figure 4B*, with two phases with rate constants of 0.45 ± 0.080 s$^{-1}$ and 0.040 ± 0.0093 s$^{-1}$ and relative amplitudes of 86 ± 2.1% and 14 ± 2.1% (*Table 3*) that are within error of those reported for *Figure 4B* (*Table 2*). Thus, dissociation of RFC and PCNA from P/T DNA is intimately coordinated such that all RFC ejects from the P/T junction simultaneously with PCNA through dissociation of both populations of the RFC•PCNA•P/T DNA complex. This is definitive proof that the clamp loading-unloading pathway reaches equilibrium after the initial loading of PCNA onto P/T DNA because subsequent loading events occur with rate constants equal to the rate constants for dissociation of RFC•PCNA from DNA. Furthermore, $A_{T,reload}/A_{T,unload}$ (0.51 ± 0.055) is equal to the fraction of labeled PCNA (0.5) in the reaction presented in *Figure 4D* (*Table 3*). This relationship holds true for other ratios of Cy5-PCNA:PCNA as well and a plot of $A_{T,Reload}/A_{T,Unload}$ vs the fraction of Cy5-PCNA in the reaction yielded a straight line with a Y-intercept of ~0.0 and a slope of ~1.0 (*Figure 4—figure supplement 3*), indicating that no PCNA is left behind on P/T DNA upon ejection of RFC. Rather, all PCNA loaded onto P/T DNA by RFC is unloaded back into solution by RFC.

The experiments depicted in *Figure 4* were also carried out on a Cy3-P/T DNA substrate lacking the single-stranded DNA flap (Cy3-P/T DNA-No Flap, *Table 1*) and yielded the same results (*Figure 4—figure supplement 4* and *Tables 2 and 3*). This demonstrates that all PCNA dissociates from the Cy3-P/T DNA substrate via the same pathway and independently of the single-stranded DNA flap and rules out the possibility that the flap traps RFC on the DNA and favors clamp unloading by keeping RFC in close proximity to the P/T junction upon loading PCNA. Indeed, human RFC is highly specific for P/T junctions and has very low affinity for purely single-stranded DNA (*Tsurimoto and Stillman, 1990*). Together, this suggests that upon loading PCNA onto a P/T junction RFC remains at or near the P/T

**Table 2.** Rate constants and relative amplitudes for the dissociation of Cy5-PCNA from Cy3-P/T DNA

| Substrate | $k_{dec,2}$ | | $k_{dec,3}$ | |
| --- | --- | --- | --- | --- |
| | Rate constant, s$^{-1}$ | % $A_T$ | Rate constant, s$^{-1}$ | % $A_T$ |
| Cy3-P/T DNA | 0.40 ± 0.083 | 83 ± 1.1 | 0.039 + 0.012 | 17 ± 1.1 |
| Cy3-P/T DNA-No Flap | 0.40 ± 0.068 | 84 ± 1.5 | 0.031 + 0.0093 | 16 ± 1.5 |
| Cy3-P/T DNA-25 bp | 0.23 ± 0.0080 | 83 ± 0.90 | 0.023 ± 0.0029 | 17 ± 0.90 |

**Table 3.** Rate constants and relative amplitudes for the dissociation of RFC from Cy3-P/T DNA

| | | $k_1$ | | $k_2$ | | |
|---|---|---|---|---|---|---|
| Substrate | polδ | Rate constant, s$^{-1}$ | % $A_T$ | Rate constant, s$^{-1}$ | % $A_T$ | $A_{T,Reload}/A_{T,Unload}$ |
| Cy3-P/T DNA | − | 0.45 ± 0.080 | 86 ± 2.1 | 0.040 ± 0.0093 | 14 ± 2.1 | 0.51 ± 0.055 |
| | + | 0.48 + 0.065 | 80 ± 2.2 | 0.042 ± 0.010 | 20 ± 2.2 | 1.0 ± 0.0063 |
| Cy3-P/T DNA-No Flap | − | 0.48 ± 0.026 | 91 ± 1.4 | 0.081 ± 0.018 | 9.0 ± 1.4 | 0.51 ± 0.028 |
| | + | 0.52 ± 0.053 | 83 ± 1.5 | 0.037 ± 0.0013 | 17 ± 1.5 | 0.99 ± 0.060 |
| Cy3-P/T DNA-25 bp | − | 0.25 ± 0.038 | 85 ± 0.64 | 0.040 ± 0.00084 | 15 ± 0.64 | 0.51 ± 0.046 |
| | + | 0.27 ± 0.012 | 70 ± 0.49 | 0.026 ± 0.0017 | 30 ± 0.49 | 1.0 ± 0.051 |

junction by anchoring to the adjacent single-stranded DNA of the template strand, not the single-stranded DNA flap. Subsequently, RFC dissociates back into solution taking all loaded PCNA with it. Thus, the opposing activities of RFC, clamp loading and unloading, both occur within the same binding encounter with a given P/T junction.

## Inhibition of the PCNA unloading activity of RFC and formation of the polymerase δ holoenzyme

Like other systems, the human replicative pols δ and ε share common binding sites on PCNA with RFC (**Zhang et al., 1999**). The results presented above imply that incoming pols must capture loaded PCNA rings from DNA-bound RFC to block unloading and complete assembly of the pol holoenzyme. In order to gain insight into how this is achieved, we repeated the trapping experiments depicted in **Figure 4C** in the presence of polδ. Under these conditions, the overall amplitude of the FRET increase ($A_{T,reload}$) will report on the amount of both unlabeled PCNA and RFC in solution as before. At a ratio of 1:1:1 PCNA:Cy5-PCNA:polδ, if polδ captures all loaded PCNA from DNA-bound RFC and all RFC ejects into solution, there will be no unlabeled PCNA in solution and RFC can only load Cy5-PCNA onto Cy3-P/T DNA. Thus, the overall amplitude of the FRET increase ($A_{T,Reload}$) should be equal to the overall amplitude of the FRET decrease ($A_{T,Unload}$) in **Figure 4B**. Furthermore, the rate of appearance of the FRET signal should be faster in the presence of polδ if polδ actively displaces RFC. As presented in **Figure 4D**, only the overall amplitude ($A_{T,Reload}$) increases in the presence of polδ. Two phases are present within the FRET time trace with rate constants and relative amplitudes that are within error of those reported in the absence of polδ (**Table 3**) as well as those reported for the dissociation of PCNA from DNA (**Table 2**). Furthermore, these values are independent of polδ concentration (**Figure 4— figure supplement 6**). Thus, polδ does not actively displace RFC from the P/T junction. Rather, RFC dissociates from the P/T DNA in a biphasic manner independently of both PCNA and polδ. Thus, the rate constants for the disappearance of the FRET signal in **Figure 4B** and re-appearance of the FRET signal in **Figure 4D** all reflect dissociation of RFC from the P/T DNA and suggest that at least two forms of RFC are present in the ternary complex. $A_{T,Reload}/A_{T,Unload}$ increased from 0.51 ± 0.055 in the absence of polδ to 1.0 ± 0.0063 in the presence of a stoichiometric amount of polδ (**Table 3**), suggesting that polδ captured all loaded PCNA from DNA-bound RFC. Importantly, this same behavior was also observed when the single-stranded flap was removed from the Cy3-P/T DNA substrate (**Figure 4—figure supplement 4B** and **Table 2**), again suggesting that RFC remains at or near P/T junction upon loading PCNA and does not retreat to the single-stranded DNA flap to allow the holoenzyme to form. When the concentration of polδ was varied, $A_{T,Reload}/A_{T,Unload}$ increased linearly to the level of the unlabeled PCNA and RFC concentrations (100 nM) and plateaued thereafter (**Figure 4—figure supplement 5**). At the break point where $A_{T,reload}/A_{T,unload}$ = 1.0, the concentrations of unlabeled PCNA, RFC, and polδ were all equivalent indicating that polδ stabilized all loaded PCNA on the Cy3-P/T DNA. Furthermore, this indicates that all PCNA loaded onto DNA by RFC is in the same form and competent for holoenzyme formation, that is the PCNA ring is closed around the P/T DNA in the correct orientation. Taken together, this suggests that polδ captures loaded PCNA rings from DNA-bound RFC to complete assembly of the pol holoenzyme. This effectively inhibits the unloading activity of DNA-bound RFC by physical occlusion and RFC subsequently dissociates from the single-stranded region of the template strand adjacent to the P/T junction, leaving behind the functional holoenzyme consisting of PCNA and polδ.

## Discussion

### RFC-catalyzed assembly of PCNA onto DNA

Previous studies on the human clamp loading-unloading pathway provided valuable insights into how this process may unfold and laid the groundwork for subsequent investigations. However, such studies were often indirect and qualitative, failing to monitor clamp loading-unloading in real time. Thus, several points along the reaction pathway have remained contested and a description of the complete reaction cycle, that is beginning and ending with RFC and PCNA free in solution, is currently lacking. By utilizing FRET experiments to monitor protein-DNA interactions, we were able to monitor the clamp loading-unloading process in real time using recombinant human proteins. Taken in a historical context, our results provide the first description of the complete human clamp loading-unloading pathway (*Figure 5*).

In the presence of ATP, RFC binds the C-terminal face of PCNA and opens the PCNA ring for assembly on P/T DNA [step 1, (*Zhang et al., 1999*; *Shiomi et al., 2000*)]. Although ring opening has yet to be directly monitored within humans, previous studies suggest that it requires only ATP binding and not hydrolysis (*Lee and Hurwitz, 1990*; *Shiomi et al., 2000*). Indeed, our pre-steady-state and steady-state FRET experiments demonstrated that assembly of the RFC•ATP•PCNA•P/T DNA complex, a process which ultimately requires PCNA ring opening, will occur in the presence of ATPγS, a non-hydrolysable analog of ATP. Once formed, the open RFC•ATP•PCNA complex specifically recognizes and binds a P/T junction [step 2, (*Tsurimoto and Stillman, 1990*; *Lee et al., 1991*)]. Our pre-steady state FRET experiments demonstrate that the bimolecular association of RFC•ATP•PCNA with DNA is very rapid, most likely diffusion-limited (*Berg and von Hippel, 1985*).

Upon binding to DNA, ATP-bound clamp loaders adopt a conserved, right-handed spiral conformation referred to as the 'notched screw cap' that closely mimics the helical geometry of the bound DNA (*Bowman et al., 2004*; *Simonetta et al., 2009*). Through extensive interactions with the C-terminal face of open clamps, clamp loaders pull the clamp subunits out-of-plane into a complementary conformation (*Miyata et al., 2005*; *Kelch et al., 2011*). Although the structure of such complexes have yet to be determined for the human system, the asymmetric DNA footprints of human RFC alone and the RFC•PCNA complex exactly agree with that predicted by crystal structures of the notched screw cap arrangement from other organisms (*Tsurimoto and Stillman, 1991*; *Bowman et al., 2004*; *Kelch et al., 2011*). Furthermore, the DNA footprints could only be mapped in the presence ATPγS, demonstrating that the notched screw cap arrangement occurs prior to ATP hydrolysis within the RFC ATPase sites. Our pre-steady state FRET experiments show that after binding P/T DNA, the RFC•ATP•PCNA•P/T DNA complex undergoes a conformational change ($k_{inc}$) that requires only the presence of ATP. Specifically, $k_{inc}$ is independent of [ATP] up to as high as 5 mM where it is assumed that each ATPase site within RFC is saturated with ATP. Furthermore, $k_{inc}$ is observed in the presence of ATPγS (*Figure 2—figure supplement 2*). Together, this suggests that $k_{inc}$ represents the RFC•ATP•PCNA complex adopting the notched screw cap arrangement on P/T DNA (step 3) in which all RFC ATPase sites are occupied by ATP.

After adopting the notched screw cap arrangement, the RFC•ATP•PCNA•P/T DNA complex undergoes a second conformational change ($k_{dec,1}$) which requires ATP hydrolysis. It should be noted that these studies cannot discern whether the observed rate constant describes the actual conformational change, a rate-limiting ATP hydrolysis, or both. In the notched screw cap arrangement, the ATPase sites of clamp loaders are aligned properly for catalysis and ATP hydrolysis is optimized (*Simonetta et al., 2009*; *Kelch et al., 2011*). Furthermore, ATP hydrolysis is essential for assembly of a PCNA clamp that can slide along DNA and associate with polδ, both processes that require closure of the PCNA ring around DNA (*Lee and Hurwitz, 1990*; *Tinker et al., 1994*). Thus, we speculate that the conformational change described by $k_{dec,1}$ reflects closing of the PCNA ring around DNA upon hydrolysis of ATP by RFC (step 4). Indeed, this conformational change is not observed in the presence of ATPγS (*Figure 2—figure supplement 2*). However, these studies cannot discern how many ATP molecules are hydrolyzed or in which ATPase sites within RFC such events occur (discussed further below).

Upon hydrolysis of ATP, RFC does not immediately eject from the PCNA•P/T DNA complex. Rather, RFC remains transiently bound near the P/T junction, dissociating slowly in a biphasic manner independently of both PCNA and polδ (*Figures 3, 4*, and *Figure 4—figure supplement 6*). Furthermore, in the absence of polδ, all PCNA loaded onto DNA by RFC is unloaded by RFC back into solution (*Figure 4*, *Table 3*, and *Figure 4—figure supplement 3*). Thus, PCNA and RFC are always present together on a given P/T DNA in the absence of polδ. However, this does not imply that RFC and PCNA remain engaged at the P/T junction while both are bound to a given DNA. In numerous reports,

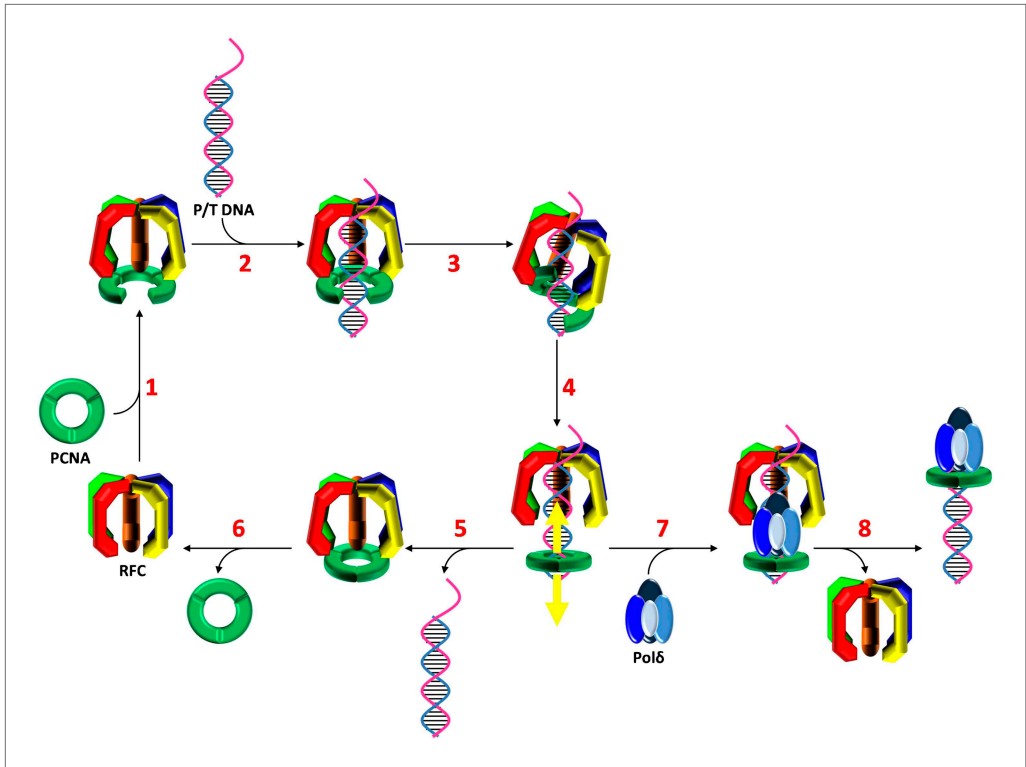

**Figure 5**. Stepwise assembly of the human DNA polymerase holoenzyme. (1) RFC•ATP binds PCNA and opens it for assembly onto DNA. (2) The open PCNA•RFC•ATP complex binds to a P/T junction and (3) adopts the notched screw cap arrangement. (4) RFC hydrolyzes ATP, closing the PCNA ring and releasing it onto DNA. (5) In the absence of polymerase, loaded PCNA is unable to 'escape' from DNA-bound RFC and is unloaded back into solution by RFC. (6) RFC subsequently releases PCNA, exchanges ADP for ATP, and the cycle repeats. (7) In the presence of polymerase, loaded PCNA is 'captured' from DNA-bound RFC by an incoming polymerase, blocking the unloading activity of DNA-bound RFC by physical occlusion. (8) RFC subsequently dissociates, leaving behind the functional holoenzyme consisting of polymerase and PCNA.

The following figure supplements are available for figure 5:

**Figure supplement 1**. The human notched screw cap complex.

**Figure supplement 2**. The clamp loader gp44/62 of T4 bacteriophage retracts towards the P/T junction upon hydrolysis of ATP and closure of the gp45 clamp ring around DNA.

PCNA•DNA complexes devoid of RFC have been isolated in vitro using recombinant human proteins. Within these studies, the circular DNA substrates did not contain any blocks to PCNA sliding (*Cai et al., 1996*, *1997*; *Yao et al., 1996*; *Uhlmann et al., 1997*; *Zhang et al., 1999*). Thus, upon hydrolysis of ATP by RFC and closure of the PCNA ring, at least some fraction of loaded PCNA must slide away from the P/T junction within these substrates, escaping the unloading activity of RFC. Indeed, RFC's affinity for PCNA decreases ~3 orders of magnitude in the presence of ADP alone (*Zhang et al., 1999*; *Shiomi et al., 2000*) and if Neutravidin is omitted from the Cy3-P/T DNA substrate, allowing PCNA to slide off the primer end of the Cy3-P/T DNA substrate, the amount of Cy5-PCNA unloaded by RFC decreases by 2.3 ± 0.27-fold (*Figure 4—figure supplement 7*). Thus, we propose that upon hydrolysis of ATP(s) and closure of the PCNA clamp ring, RFC releases PCNA onto DNA (step 4). This would also be most consistent with the passive model for the capture of loaded PCNA by polδ (discussed later).

In the absence of polδ, all loaded PCNA re-engages with DNA-bound RFC and the RFC•PCNA complex together ejects from the P/T DNA into solution (step 5). This implies that RFC must re-open the closed PCNA ring prior to unloading. Based on the requirements for PCNA ring opening in solution, it is presumed that only ATP binding, not hydrolysis, is required for opening PCNA on DNA. We are

currently testing this hypothesis and thus it cannot be ruled out that RFC requires the energy of ATP hydrolysis to pry open closed PCNA rings on DNA that are reinforced by electrostatic interactions between the positively-charged inner surface of PCNA and the negatively-charged DNA backbone. Regardless, unloading of PCNA from DNA is not limited by re-opening of the PCNA ring. Rather, this process is entirely rate-limited by RFC dissociation from P/T DNA as the rate constants for dissociation of RFC•PCNA and RFC alone are equivalent (*Figure 4* and *Table 3*). Dissociation of RFC from P/T DNA is biphasic, with ~80% dissociating fast and the remaining 20% dissociating ~10-fold slower. This suggests that at least two forms of RFC are present with only the faster of the two phases being dependent upon ATP hydrolysis. As discussed above, clamp loaders revert to a low-affinity DNA-binding state upon hydrolysis of ATP and eject from the P/T junction. Perhaps as the concentration of ATP is increased, nucleotide exchange begins to compete with ATP hydrolysis and/or ADP binding such that dissociation of most RFC from the P/T junction is inhibited. This suggests that the minor population of RFC represents that which dissociates from the P/T junction independently of its nucleotide-bound state. Once free in solution, RFC releases bound PCNA, exchanges ADP for ATP (step 6), and the cycle repeats.

The proposed model suggests that there are at least two distinct ATP hydrolysis-dependent steps that occur within the initial binding encounter of RFC with a P/T junction. This implies that ATP hydrolysis occurs 'all or none' or sequentially. In the former model, each ATP-hydrolysis dependent event is catalyzed by ATP hydrolysis occurring within all ATPase sites of RFC at once. Thus, in our PCNA loading-unloading model, nucleotide exchange would have to occur while RFC is still bound to P/T DNA, in between closure of the clamp ring and unloading of PCNA into solution. In the latter model, there is a division of labor in which the ATPase sites of certain RFC subunits 'fire' at defined moments along the reaction coordinate in order to catalyze distinct steps (i.e., ring closure, ejection of the clamp loader, etc.). Although our studies on human RFC cannot discern between these models, previous reports on clamp loaders from *E. coli* (*Snyder et al., 2004*), the hyperthermophilic euryarchaeon *Archaeoglobus fulgidus* (*Seybert and Wigley, 2004*), and *S. cerevisiae* (*Johnson et al., 2006*) have provided experimental evidence for the latter model and recent crystallographic studies on the T4 bacteriophage clamp loader have provided the first structural insights. In brief, after the clamp loader•clamp complex binds to a P/T junction, ATP is hydrolyzed in one of the clamp loader subunits causing the open clamp to close around the DNA. The conformational changes within the hydrolyzing subunit are propagated to the remaining subunits of the clamp loader, presumably inducing sequential ATP hydrolysis (*Kelch et al., 2011*). Thus, the state of bound nucleotide within the clamp loader progressively changes along the reaction coordinate, temporally defining the catalytic events. Future studies with human proteins will discern whether such a model is conserved.

## Formation of the replicative polymerase holoenzyme

As discussed above, in the absence of a replicative pol, RFC loads a stoichiometric amount of PCNA onto a P/T junction and then disengages, taking all loaded PCNA along with it. The results presented in *Figure 4* and *Table 3* demonstrate that polδ captures loaded PCNA from DNA-bound RFC, inhibiting RFC's unloading activity by physical occlusion and completing assembly of the pol holoenzyme. Furthermore, polδ stabilizes a stoichiometric amount of loaded PCNA on DNA demonstrating that all loaded PCNA is closed around the P/T DNA and competent for holoenzyme assembly (*Figure 4—figure supplement 5*). This high efficiency is in contrast to holoenzyme assembly in *S. cerevisiae* where only one of the two populations of loaded PCNA is competent for polδ binding (*Kumar et al., 2010*).

The human replicative pols (δ and ε) share common binding sites on PCNA with RFC (*Zhang et al., 1999*). This suggests that assembly of the human holoenzyme may occur by one of two models. Incoming pols may either actively displace RFC from an RFC•PCNA complex present at the P/T junction or passively capture a closed PCNA clamp while it is transiently disengaged from the DNA-bound RFC. As presented in *Table 3*, and *Figure 4—figure supplement 6*, dissociation of RFC from the P/T DNA is independent of polδ, consistent with a passive capture of loaded PCNA. This agrees with the data presented here and elsewhere that suggest that PCNA is released onto DNA upon hydrolysis of ATP by RFC (*Cai et al., 1996*, *1997*; *Yao et al., 1996*; *Uhlmann et al., 1997*; *Zhang et al., 1999*). However, it should be noted that this does not rule out that DNA-bound RFC may 'chaperone' in replicative pols to the P/T junction as suggested to occur during nucleotide excision repair (*Overmeer et al., 2010*). Indeed, our studies suggest that RFC remains transiently bound near the P/T DNA upon binding of polδ to the loaded PCNA clamp (discussed further below) and human RFC has been shown to interact with polδ (*Yuzhakov et al., 1999*).

In the notched screw cap complex, human RFC anchors to the single-stranded region of the P/T junction, covering 12 nucleotides of the template strand, and extends into the double-stranded region, covering 15 and 8 nucleotides within the primer and template strands, respectively (*Tsurimoto and Stillman, 1991*; *Gulbis et al., 1996*). The open PCNA clamp expands the protected region on the double-stranded side of the P/T junction out to ~25 bp (*Figure 5—figure supplement 1*). This arrangement of the clamp loader•ATP•clamp complex is highly conserved in all domains of life as well as T4 bacteriophage (*Hedglin et al., in press*) as discussed further below. The closed ring of human PCNA has a width (30 Å) that is equivalent to ~10 bp of B-form DNA (*Gulbis et al., 1996*). Thus, due to the minimal double-stranded region of the P/T junction (30 bp) and the Neutravidin block within the Cy3-P/T DNA substrate, loaded PCNA can slide at most 20 bp away from the P/T junction to allow an incoming polymerase to bind to its C-terminal face (*Figure 1A*). We believe this best represents the situation in vivo where PCNA is loaded onto ~30 nt RNA-DNA hybrid primers that are blocked on both sides by single-stranded DNA binding protein (replication protein A, RPA, in humans) that restrict the loaded PCNA to the P/T junction (*Burgers, 2009*). However, the experiments performed on the Cy3-P/T DNA substrate lacking the single-stranded DNA flap (*Figure 4—figure supplement 4* and *Table 3*) demonstrate that human RFC remains at or near the P/T junction upon loading PCNA onto DNA by anchoring to the adjacent single-stranded DNA of the template strand, implying that RFC must also retract towards the P/T junction upon closure and release of PCNA onto DNA to allow for an incoming DNA polymerase. Recent, ground-breaking crystallographic studies from the Kuriyan laboratory suggest this may be the case. In their 2011 publication, Kelch et al. crystallized the clamp loader•clamp•DNA complex from T4 bacteriophage in two forms (*Figure 5—figure supplement 2*). In the ATP bound form, the clamp loader•clamp complex has adopted a right-handed spiral conformation (notched screw cap) that closely mimics the helical geometry of the bound DNA. In another form, the clamp is closed around DNA and the B subunit of the clamp loader has hydrolyzed ATP, moved away from the adjacent C subunit, and partially disengaged from the template strand and the clamp. The new conformation of the B subunit is incompatible with the symmetric spiral of the other clamp loader subunits. This partial collapse of the clamp loader spiral retracts the clamp loader, particularly the N-terminal domain of the A subunit, towards the P/T junction. The authors suggest that as ATP hydrolysis continues around the clamp loader spiral, the matching symmetry between the clamp loader and the DNA•clamp is progressively broken (*Kelch et al., 2011*). Perhaps as ATP hydrolysis propagates around the human clamp loader after closing of the clamp ring on DNA, RFC retracts towards the 3' end of the primer strand as the clamp loader spiral collapses, revealing more of the double-stranded region of the P/T junction. In addition to the closed PCNA ring sliding away from the P/T junction, such a conformational change would allow sufficient room for an incoming polymerase to form the holoenzyme. Indeed, limiting the double-stranded region of the P/T junction to the DNA footprint (25 bp) of the human notched screw cap complex (Cy3-P/T DNA-25 bp in *Table 1*) had no effect on the amount of PCNA unloaded in the absence of polδ or the amount of polδ holoenzyme formed (*Figure 4—figure supplement 8* and *Tables 2 and 3*), suggesting that DNA-bound RFC retracts towards the P/T junction upon closure and release of PCNA onto DNA (*Figure 5*, step 4). Once loaded PCNA is captured by polδ (*Figure 5*, step 7), RFC subsequently dissociates, leaving behind the functional holoenzyme consisting only of PCNA and polδ (*Figure 5*, step 8).

## Unloading of sliding clamps during S-phase

The amount of PCNA trimers present in the nucleus of human cells during S-phase is estimated to be ~$2 \times 10^5$/cell (*Morris and Mathews, 1989*). Based on the size of the human genome and the number of P/T junctions (Okazaki fragments and origins of replication), it is suggested that PCNA trimers are in demand during S-phase and must be re-used hundreds to thousands of times per cell cycle. As discussed above, human PCNA is incredibly stable on DNA in the absence of free DNA ends and an enzymatic unloading activity. Thus, an efficient unloading mechanism is required for recycling during S-phase (*Yao et al., 1996*; *Leu et al., 2000*). The results presented here demonstrate that RFC loads PCNA onto a P/T junction and, in the absence of polδ, catalytically unloads all loaded PCNA from the P/T DNA without first dissociating into solution. This may serve to maximize the utilization of limited PCNA by inhibiting the build-up of free PCNA on DNA in the absence of polymerase. Presumably, once polδ disengages from the P/T junction and PCNA, RFC may unload the PCNA from DNA, recycling limited PCNA to keep up with ongoing replication. However, in replicating eukaryotic cells, PCNA left behind on DNA serves as a cell signal during S-phase by marking replicated DNA for

chromatin assembly (*Shibahara and Stillman, 1999*) and sister chromatid cohesion (*Moldovan et al., 2006*). Furthermore, replication-blocking lesions encountered during S-phase are also marked by PCNA for replication by alternative DNA polymerases that can accommodate lesions. Such a process, referred to as translesion (TLS) synthesis, is initiated by ubiquitylation of PCNA at the site of the lesion and may occur immediately at the replication fork or at a later time (*Sale et al., 2012*). Thus, some PCNA rings must remain on DNA after dissociation of replicative pols. This suggests that the PCNA unloading activity of RFC is tightly coordinated with the aforementioned activities, perhaps through post-translational modifications of PCNA, RFC, or both. Indeed, SUMO (small ubiquitin-like modifier) is attached to PCNA during S-phase in *S. cerevisiae*. This serves to prevent homologous recombination by recruiting the Srs2 helicase but also leads to accumulation of SUMO-PCNA on chromatin in the absence of an alternative clamp loader, ELG1-RFC, suggesting that this modification may also inhibit the PCNA unloading activity of native RFC (*Parnas et al., 2010*). Attachment of SUMO to PCNA has only recently been established in humans and it remains to be seen what effect, if any, this modification has on the stability of the human PCNA clamp on DNA (*Gali et al., 2012*; *Moldovan et al., 2012*). Furthermore, it was recently shown that human RFC is ubiquitylated in a DNA damage-dependent manner in vivo (*Tomida et al., 2008*). Although it is currently unknown what effect this modification has on the catalytic activities of RFC, its temporal correlation with ubiquitylation of PCNA and TLS tempts one to speculate that ubiquitin conjugation to RFC may serve to inhibit its PCNA unloading activity during TLS.

## Materials and methods

### Oligonucleotides and recombinant proteins

DNA constructs were synthesized by Integrated DNA Technologies (Coralville, IA) and purified on denaturing polyacrylamide gels. Concentrations were determined from the absorbance at 260 nm using the calculated extinction coefficients. For annealing of all substrates, the Cy3-labeled primer strand was mixed with a 1.1-fold excess each of Biotin-labeled template and flap in 1× Annealing Buffer (10 mM TrisHCl, pH 8.0, 100 mM NaCl, 1 mM EDTA), heated to 95°C for 5 min, and allowed to slowly cool to room temperature. The sequences of the primer, template, and flap constructs for each forked DNA substrate are shown in *Table 1*. In all experiments described in this study, DNA was bound to a fourfold excess of Neutravidin (Sigma-Aldrich) unless otherwise stated.

The plasmid for expression of wild-type human RFC was a generous gift from Dr. Paul Modrich (Duke University School of Medicine, Durham, NC). Details of the truncated human RFC expression vectors and purification procedures will be described elsewhere (manuscript in preparation). Wild-type human polymerase δ (polδ) was expressed and purified from *E. coli* as described previously (*Masuda et al., 2007*). RFC and polδ concentrations were determined via Bradford Assay using BSA as a standard. The plasmid for expression of poly(His)-tagged wild-type PCNA was a generous gift from Prof. Ulrich Hubscher (University of Zurich Institute of Veterinary Biochemistry and Molecular Biology, Zurich, Switzerland). Details of the human PCNA mutant (mutPCNA) expression vector will be described elsewhere (manuscript in preparation). Wild-type and mutant PCNA were expressed in *E. coli* and purified by a published protocol (*Jonsson et al., 1995*). PCNA concentrations were determined from the absorbance at 280 nm using the calculated extinction coefficients.

### Labeling of mutant PCNA

The mutant human PCNA was labeled with Cy5-maleimide at N107C and checked for activity as described previously for *S. cerevisae* PCNA (*Kumar et al., 2010*). The extent of cysteine-labeling was determined by calculating the molar concentrations of the dye ($\varepsilon_{650} = 250,000$ M$^{-1}$cm$^{-1}$) and the protein ($\varepsilon_{280} = 13,670$ M$^{-1}$cm$^{-1}$) by absorbance and correcting for the absorbance of the dye at 280 nm (~5% of the absorbance at 650 nm). PCNA monomer labeling efficiency was 38.3%, indicating that each PCNA trimer has at least one labeled monomer on average.

### Steady-state ATPase activity

ATPase activity of RFC (hRFCp140ΔN555) was assayed spectrophotometrically via an NADH oxidation enzyme-coupled assay as previously described (*Norby, 1988*; *Kumar et al., 2010*). The ATPase activity was determined at 25°C in an assay solution containing 1× Replication Buffer (25 mM TrisOAc, pH 7.5, 10 mM Mg(OAc)$_2$, 125 mM KOAc), 0.1 mg/ml BSA, 1 mM DTT, 1 mM ATP, 125 nM RFC, 125 nM forked DNA (500 nM Neutravidin), and 125 nM of either wild-type PCNA (wtPCNA), mutant PCNA (mutPCNA), or Cy5-labeled mutant PCNA (Cy5-PCNA). The initial rates of ATP hydrolysis are reported.

## Steady-state fluorescence

Steady-state fluorescence spectroscopy was carried out in a Jobin Yvon FluoroMax-4 fluorimeter. The buffer solution contained 1× Replication Buffer, 0.1 mg/ml BSA, and 1 mM DTT. The assay solution contained 250 nM forked DNA, 1 µM Neutravidin, and 5 mM of either ATP or ATPγS equilibrated at 25°C. To this solution, 250 nM Cy5-labeled PCNA mutant and 250 nM RFC were sequentially added and fluorescence emission spectra recorded from 530 to 725 nm after 514 nm excitation.

## Stopped-flow fluorescence spectroscopy

Stopped-flow studies were performed on an Applied Photophysics SX20 stopped-flow machine equipped with a fluorescence detector. PCNA loading experiments were performed by mixing RFC•Cy5-PCNA•ATP in one syringe with Cy3-P/T DNA in the other syringe, unless stated otherwise. PCNA unloading experiments were performed by mixing RFC•Cy5-PCNA•ATP•Cy3-P/T DNA in one syringe with unlabeled mutant PCNA in the other syringe, unless stated otherwise. Loading and unloading was monitored by exciting the donor (Cy3) at 514 nm and following the resulting FRET signal using a 645 nm cutoff filter (Andover Corporation, Salem, NH). The FRET traces for the PCNA loading experiments were recorded over 60 s by collecting 8000 total time points; 5000 time points over the initial 10 s of the reaction and 3000 time points over the remaining 50 s. The FRET traces for the PCNA unloading experiments were recorded over 60–180 s by collecting 1000 time points per 60 s. All traces were analyzed using Applied Photophysics ProData™ software. Conditions for each experiment are detailed in the figure captions.

For the PCNA loading experiments presented in **Figure 2**, the FRET signal remains flat in the absence of RFC and represents the zero FRET state (i.e., fluorescence value at zero time, $y_0$). However, in the presence of RFC, an initial FRET increase occurs within the dead time of the instrument such that the observed FRET value is elevated at zero time. Although this phase cannot be fit to a rate equation, its amplitude ($A_{dead}$) is reflected in the fluorescence value at infinite time (c) and can be deduced by modifying the double exponential equation as follows. For a given condition

$$y = A_{inc}e^{\wedge}(-k_{inc}t) + A_{dec,1}e^{\wedge}(-k_{dec,1}t) + c', \text{ where } c' = y_0 + A_{dead} + A_{inc} + A_{dec,1}, \quad (1)$$

where y represents the fluorescence value at time t, $A_{dead}$ the amplitude of the initial FRET increase occurring within the dead time of the instrument, $A_{inc}$ and $k_{inc}$ the amplitude and rate constant for the observed FRET increase, $A_{dec,1}$ and $k_{dec,1}$ the amplitude and rate constant for the observed FRET decrease, and c' the fluorescence value at infinite time. $y_0$ represents the fluorescence value at time 0 and was obtained at 0 nM RFC (**Figure 2C**) where the FRET signal remains flat over the entire time course. Thus, for a given condition,

$$A_{dead} = c' - y_0 - A_{inc} - A_{dec,1}. \quad (2)$$

## Acknowledgements

We would like to thank Dr. Vladimir Bermudez and Inger Tappin from the Hurwitz laboratory at the Memorial Sloan Kettering Cancer Center for helpful discussions on the purification, stability, and storage of human RFC.

## Additional information

### Funding

| Funder | Grant reference number | Author |
| --- | --- | --- |
| National Institutes of Health | GM13306 | Stephen Benkovic |
| National Institutes of Health NRSA Fellowship | F32CA165471 | Mark Hedglin |

The funders had no role in study design, data collection and interpretation, or the decision to submit the work for publication.

## Author contributions

MH, Conception and design, Acquisition of data, Analysis and interpretation of data, Drafting or revising the article; SKP, Acquisition of data, Drafting or revising the article, Contributed unpublished essential data or reagents; ZH, Drafting or revising the article, Contributed unpublished essential data or reagents; SB, Conception and design, Analysis and interpretation of data, Drafting or revising the article

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
