## [Decision Letter]

Thank you for choosing to send your work entitled “Stepwise Assembly of the Human Replicative Polymerase Holoenzyme” for consideration at *eLife*. Your article has been reviewed in depth by a Senior editor (John Kuriyan) and 2 other reviewers. The Senior editor and the other reviewers discussed their critiques and the Senior editor has assembled the following questions and suggestions based on the online discussion.

In this paper Benkovic and colleagues analyze the interaction between the DNA polymerase clamp, PCNA, the clamp loader (RFC), and primer-template DNA. The manuscript describes a series of carefully designed and closely argued FRET experiments that seek to dissect the detailed conformational steps in the loading and unloading of human PCNA by human RFC. The advance reported in this paper concerns the role of the RFC complex in unloading PCNA from DNA after it has been loaded.

As the authors point out, it is well understood that PCNA does not readily load on to DNA on its own, and that once loaded it is very stable on DNA because it forms a closed ring. It is also appreciated, from previous work, that clamp loaders can unload clamps from DNA, in the reverse of the reaction that puts them on to DNA, which helps recycling clamps from Okazaki fragments.

Hedglin et al. now present a kinetic analysis of human RFC focused on both the clamp loading and unloading phases of the reaction. The authors delineate distinct ATP binding and ATP hydrolysis steps in the loading reaction, and show that PCNA transiently released from RFC following ATP hydrolysis can be recaptured by RFC in the first step of an unloading reaction. At one level, this is not surprising, and reflects what would be predicted by micro-reversibility of an overall equilibrium process. Nevertheless, the major conclusion, which is that RFC does not dissociate after loading PCNA and can unload the clamp from DNA, runs counter to the prevailing view and therefore increases interest in the paper. The authors suggest that RFC efficiently loads and unloads clamps at primed DNA within a single binding encounter with the primer-template junction. This has major biological implications as it would lower the efficiency of clamp loading during DNA replication, when the required loading rate is ∼1 clamp/second. The problem would be acute if there were any lag between clamp loading and polymerase arrival. Thus, the conclusions of this study need to be carefully validated because they have the potential to change our understanding of a fundamental aspect of this important step in DNA replication.

There are two major concerns about the generality of the conclusions drawn from the experiments reported in this paper, both of which have to do with the nature of the DNA construct used. The experiments are well performed, and the reasoning behind the experiments is well explained, but the experiments all rely on a particular DNA construct, shown in Figure 1. This construct is designed to block PCNA sliding off the DNA in two ways. First, streptavidin is bound to a biotin tag on one end. Second, a single-stranded overhang is present at the other end, which also provides a block for PCNA sliding. The two major concerns are as follows:

1) Various clamp loaders, such as *E. coli* γ complex, as well as human and yeast RFC are known to bind single stranded DNA with high affinity. Does the 5' ssDNA flap shift the equilibrium towards clamp unloading by providing a binding site for RFC and enabling it to remain close to loaded PCNA? If there were no such extra binding sites for RFC (no ssDNA in the substrate or the presence of RPA), would there be any significant clamp unloading in a single cycle?

2) From a large body of previously published work we know that polymerases and clamp loaders compete for the same binding sites on the clamp, and so it is not surprising that the arrival of the polymerase (Polδ is used in the experiments reported here) prevents RFC from unloading PCNA. But, in the context of the experiments described here, there is difficulty in understanding how Polδ accesses the same face of the PCNA clamp as does RFC, if a transient association between RFC and PCNA is being maintained. There is a concern that the distance between the RFC loading site and the biotin-avidin block used in this study overly restricts the degree to which the released PCNA can truly dissociate from RFC. Is RFC moving away from the double-helical region and being held by the single-stranded DNA, as discussed in point 1, above? If so, would such an interaction be physiologically relevant?

In light of these two concerns, the paper would be greatly strengthened if the authors explored (A) the effect of removing the single-stranded overhang or the addition of single-stranded DNA binding protein, e.g., RPA and (B) the effect of longer spacings between the P/T junction and the biotin-avidin block to properly define the extent of dissociation that is needed for Polδ to intervene, and the effect of PCNA diffusion away from the loading site on the kinetics of re-binding to RFC in the unloading process. In the absence of experiments on variants of the DNA construct used, there would be lingering questions about whether the results are an artifact of RFC and PCNA being trapped by a DNA construct that differs in important ways from the structure of a replication fork. We recognize that certain aspects of the DNA construct, such as the flap, may not easily be changed, but if these concerns cannot be overcome then the findings could be open to alternative interpretations.

---

## [Author Response]

*1) Various clamp loaders, such as* E. coli *γ complex, as well as human and yeast RFC are known to bind single stranded DNA with high affinity. Does the 5' ssDNA flap shift the equilibrium towards clamp unloading by providing a binding site for RFC and enabling it to remain close to loaded PCNA? If there were no such extra binding sites for RFC (no ssDNA in the substrate or the presence of RPA), would there be any significant clamp unloading in a single cycle*?

We removed the 17 nt single-stranded DNA flap from the Cy3-P/T DNA substrate and did not observe any effect on dissociation of Cy5-PCNA from Cy3-P/T DNA or formation of the polδ holoenzyme (Figure 4–figure supplement 4 and Tables 1 and 2). More details and further discussion has been added to the main text. This rules out the possibility that the flap traps RFC on the DNA and favors clamp unloading by keeping RFC in close proximity to the P/T junction upon loading PCNA.

*2) From a large body of previously published work we know that polymerases and clamp loaders compete for the same binding sites on the clamp, and so it is not surprising that the arrival of the polymerase (Polδ is used in the experiments reported here) prevents RFC from unloading PCNA. But, in the context of the experiments described here, there is difficulty in understanding how Polδ accesses the same face of the PCNA clamp as does RFC, if a transient association between RFC and PCNA is being maintained. There is a concern that the distance between the RFC loading site and the biotin-avidin block used in this study overly restricts the degree to which the released PCNA can truly dissociate from RFC. Is RFC moving away from the double-helical region and being held by the single-stranded DNA, as discussed in point 1, above? If so, would such an interaction be physiologically relevant*?

Our apologies if we were unclear in the description of our model. Our model does not propose that RFC and PCNA remain engaged at the P/T junction while both are bound to a given DNA. The results presented in Figure 4–figure supplement 7, along with previously published reports, demonstrates that upon hydrolysis of ATP and closure of the clamp ring, RFC releases PCNA onto DNA. Due to the minimal double-stranded region of the P/T junction (30 bp) and the Neutravidin block within the Cy3-P/T DNA substrate, loaded PCNA can slide at most 20 base pairs away from the P/T junction to allow an incoming polymerase to bind to its C-terminal face. We believe this best represents the situation *in vivo* where PCNA is loaded onto ∼30 nt RNA-DNA hybrid primers that are blocked on both sides by RPA that restricts loaded PCNA to the P/T junction.

The experiments performed on the Cy3-P/T DNA substrate lacking the single-stranded DNA flap (Figure 4–figure supplement 4 and Table 2) demonstrate that human RFC does not retreat to the single-stranded flap upon loading PCNA onto the Cy3-P/T DNA substrate. Rather, the results from these studies suggest that RFC remains at or near P/T junction by anchoring to the adjacent single-stranded DNA of the template strand. Please see the main text for more details and further discussion. In the experiments presented in Figure 4–figure supplement 8 and Table 2, we limited the double-stranded region of the P/T junction to the DNA footprint of the “notched screw cap” conformation of the human RFC•ATP•PCNA complex (25 bp) and did not observe any effect on the amount of polδ holoenzyme formed. Please see the main text for more details and further discussion.

This suggests that, in addition to the closed PCNA ring sliding away from the P/T junction, DNA-bound RFC also retracts towards the P/T junction upon closure and release of PCNA onto DNA. Together, this allows sufficient room for an incoming polymerase to bind to the C-terminal face of PCNA and access the 3' end of the primer strand to form the holoenzyme. This is in agreement with recent structural studies from T4 bacteriophage that showed that hydrolysis of ATP within one subunit of the T4 clamp loader gp44/62 causes the open gp45 sliding clamp to close around DNA, and the clamp loader spiral to partially collapse and retract towards the P/T junction.

*In the absence of experiments on variants of the DNA construct used, there would be lingering questions about whether the results are an artifact of RFC and PCNA being trapped by a DNA construct that differs in important ways from the structure of a replication fork*.

The two major findings of our studies were as follows:

1) In the absence of a DNA polymerase, the two opposing activities of RFC, clamp loading and unloading, occur within the same binding encounter with a given DNA. Specifically, RFC loads a stoichiometric amount of PCNA onto a P/T junction and then dissociates into solution, taking all loaded PCNA along with it.

2) Polδ passively captures DNA-bound PCNA and physically occludes it from the unloading activity of RFC in order to form the replicative polymerase holoenzyme. RFC subsequently dissociates, leaving behind the holoenzyme consisting of only PCNA and polδ.

These findings were both independently supported by experiments performed on three different DNA constructs (please see Table 2 for reference), strongly suggesting that these results are not an artifact of the DNA construct.